# Efficient Training-Free High-Resolution Synthesis with Energy Rectification in Diffusion Models

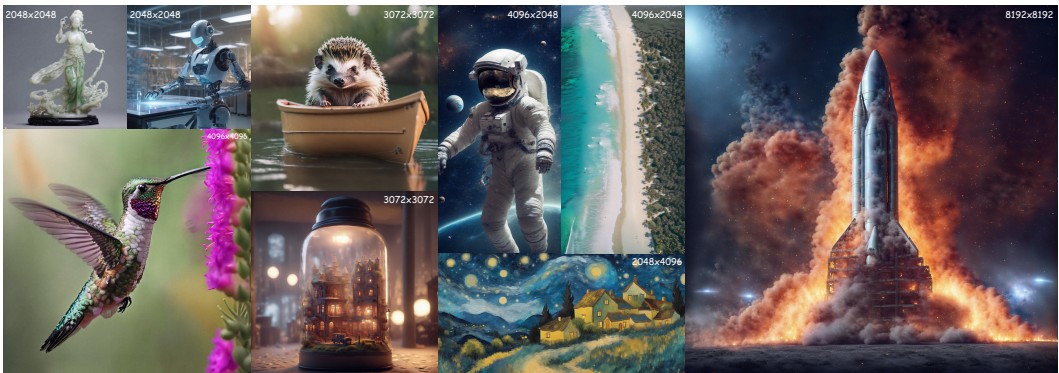

Figure 1: Generated images by *RectifiedHR*. The training-free *RectifiedHR* enables diffusion models (SDXL is shown in the figure) to synthesize images at resolutions exceeding their original training resolution. Please zoom in for a closer view.

## Abstract

Diffusion models have achieved remarkable progress across various visual generation tasks. However, their performance significantly declines when generating content at resolutions higher than those used during training. Although numerous methods have been proposed to enable high-resolution generation, they all suffer from inefficiency. In this paper, we propose *RectifiedHR*, a straightforward and efficient solution for training-free high-resolution synthesis. Specifically, we propose a noise refresh strategy that unlocks the model's training-free high-resolution synthesis capability and improves efficiency. Additionally, we are the first to observe the phenomenon of energy decay, which cause image blurriness during the high-resolution synthesis process. To address this issue, we introduce average latent energy analysis and find that tuning the classifier-free guidance hyperparameter can significantly improve generation performance. Our method is entirely training-free and demonstrates efficient performance. Furthermore, we show that *RectifiedHR* is compatible with various diffusion model techniques, enabling advanced features such as image editing, customized generation, and video synthesis. Extensive comparisons with numerous baseline methods validate the superior effectiveness and efficiency of *RectifiedHR*.

## 1 Introduction

Recent advances in diffusion models (Rombach et al., 2022; Podell et al., 2023; Chen et al., 2023b; Li et al., 2024b; Zhuo et al., 2024; Labs, 2023; Esser et al., 2024; Luo et al., 2023; Liu et al., 2024a) have significantly improved generation quality, enabling realistic editing (Yang et al., 2023; Miyake et al., 2023; Tumanyan et al., 2023; Brooks et al., 2023; Bar-Tal et al., 2022; Couairon et al., 2022; Kawar et al., 2023; Mokady et al., 2023) and customized generation (Li et al., 2024a; Bar-Tal et al.,

Figure 2: The visualization images corresponding to "predicted $x_0$" at different time step t, abbreviated as $p_{x_0}^t$. The figure visualizes the process of how $p_{x_0}^t$ changes with the sampling steps, where the x-axis represents the timestep in the sampling process. The 11 images are evenly extracted from 50 steps. Early steps primarily establish global structure, while later steps refine local details; toward the end, $p_{x_0}^t$ exhibits RGB-like characteristics.

2023; Tewel et al., 2023; Gal et al., 2022; Ruiz et al., 2023b; Ding et al., 2024). However, these models struggle to generate images at resolutions beyond those seen during training, resulting in noticeable performance degradation. Training directly on high-resolution content is computationally expensive, underscoring the need for methods that enhance resolution without requiring additional training.

Currently, the naive approach is to directly input high-resolution noise. However, this method leads to severe repeated pattern issues. To address this problem, many training-free high-resolution generation methods have been proposed, such as (Bar-Tal et al., 2023; Lee et al., 2023; Du et al., 2024; Lin et al., 2025; 2024; He et al., 2023; Huang et al., 2025; Zhang et al., 2023b; Jin et al., 2023; Hwang et al., 2024; Haji-Ali et al., 2024; Shi et al., 2024; Liu et al., 2024b; Kim et al., 2024; Cao et al., 2024; Zhang et al., 2024; Guo et al., 2024; Wu et al., 2024). However, these methods all share a common problem: they inevitably introduce additional computational overhead. For example, the sliding window operations introduced by (Bar-Tal et al., 2023; Lee et al., 2023; Du et al., 2024; Lin et al., 2025; 2024; Hwang et al., 2024) have overlapping regions that result in redundant computations. Similarly, (Shi et al., 2024; Liu et al., 2024b; Lin et al., 2025) require setting different prompts for small local regions of each image and need to incorporate a vision-language model. Additionally, (Kim et al., 2024; Cao et al., 2024; Zhang et al., 2024) require multiple rounds of SDEdit (Meng et al., 2021) or complex classifier-free guidance (CFG) to gradually increase the resolution from a low-resolution image to a high-resolution image, thereby introducing more sampling steps or complex CFG calculations. All of these methods introduce additional computational overhead and complexity, significantly reducing the speed of high-resolution synthesis.

We propose an efficient framework, *RectifiedHR*, to enable high-resolution synthesis by progressively increasing resolution during sampling. The simplest baseline is to progressively increase the resolution in the latent space. However, naive resizing in latent space introduces noise and artifacts. We identify two critical issues and propose corresponding solutions: (1) Since the latent space is obtained by transforming RGB images via a VAE, RGB-based resizing becomes invalid in the latent space (Tab. 2, Method D). Moreover, as the latent comprises "predicted $x_0$" and Gaussian noise, direct resizing distorts the noise distribution. To address this, we propose noise refresh, which independently resizes "predicted $x_0$"—shown to exhibit RGB characteristics in late sampling (Fig. 2)—and injects fresh noise to maintain a valid latent distribution while increasing resolution. (2) We are the first to observe that resizing "predicted $x_0$": introduces spatial correlations, reducing pixel-wise independence, causing detail loss and blur, and leading to energy decay (Fig. 3a). To mitigate this, we propose energy rectification, which adjusts the CFG hyperparameter (Fig. 3b) to compensate for the energy decay and effectively eliminate blur. Compared to (Kim et al., 2024; Cao et al., 2024; Zhang et al., 2024), our method achieves high-resolution synthesis without additional sampling steps or complex CFG calculations, ensuring computational efficiency.

In general, our main contributions are as follows: (1) We propose *RectifiedHR*, an efficient, training-free framework for high-resolution synthesis that eliminates redundant computation and enables resolution scalability without requiring additional sampling steps. (2) We introduce noise refresh and energy rectification, pioneering the use of average latent energy analysis to address energy decay—an issue previously overlooked in high-resolution synthesis. (3) Our method surpasses existing baselines in both efficiency and quality, achieving faster inference while preserving superior fidelity. (4) We demonstrate that *RectifiedHR* can be seamlessly integrated with ControlNet, supporting a range of applications such as image editing, customized image generation, and video synthesis.

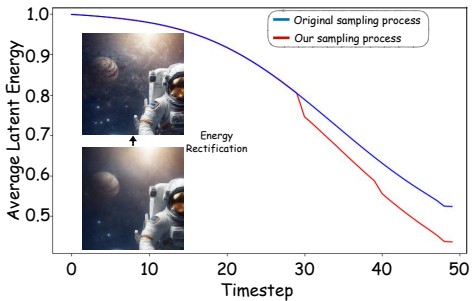 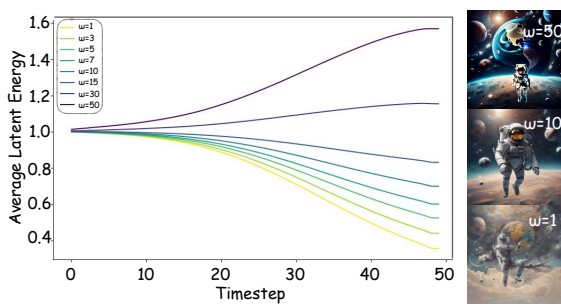

(a) The energy decay phenomenon of our noise refresh sampling process is evaluated in comparison to the original sampling process across 100 random prompts.

(b) The evolution of average latent energy over timesteps during the generation of $1024 \times 1024$ resolution images from 100 random prompts under different classifier-free guidance hyperparameters.

Figure 3: (a) The x-axis denotes the timesteps of the sampling process, and the y-axis indicates the average latent energy. The blue line shows the average latent energy of the original sampling process when generating $1024 \times 1024$-resolution images. The red line corresponds to our noise refresh sampling process, where noise refresh is applied at the 30th and 40th timesteps, and the resolution progressively increases from $1024 \times 1024$ to $2048 \times 2048$, and subsequently to $3072 \times 3072$. It can be observed that noise refresh induces a noticeable decay in average latent energy. From the left images, it is evident that after energy rectification, image details become more pronounced. (b) The x-axis represents the timestep, the y-axis represents the average latent energy, and $\omega$ denotes the hyperparameter for classifier-free guidance. It can be observed that the average latent energy increases as $\omega$ increases. From the right figures, one can observe how the generated images vary with increasing $\omega$.

## 2 RELATED WORK

### 2.1 TEXT-GUIDED IMAGE GENERATION

With the scaling of models, data volume, and computational resources, text-guided image generation has witnessed unprecedented advancements, leading to the emergence of numerous diffusion models such as LDM (Rombach et al., 2022), SDXL (Podell et al., 2023), PixArt (Chen et al., 2023b; 2025), HunyuanDiT (Li et al., 2024b), LuminaNext (Zhuo et al., 2024), FLUX (Labs, 2023), SD3 (Esser et al., 2024), LCM (Luo et al., 2023), and UltraPixel (Ren et al., 2024). These models learn mappings from Gaussian noise to high-quality images through diverse training and sampling strategies, including DDPM (Ho et al., 2020), SGM (Song et al., 2020b), EDM (Karras et al., 2022), DDIM (Song et al., 2020a), flow matching (Lipman et al., 2022), rectified flow (Liu et al., 2022), RDM (Teng et al., 2023), and pyramidal flow (Jin et al., 2024). However, these methods typically require retraining and access to high-resolution datasets to support high-resolution generation. Consequently, exploring training-free approaches for high-resolution synthesis has become a key area of interest within the vision generation community. Our method is primarily designed to enable efficient, training-free high-resolution synthesis in a plug-and-play manner.

### 2.2 TRAINING-FREE HIGH-RESOLUTION IMAGE GENERATION

Due to the domain gap across different resolutions, directly applying diffusion models to high-resolution image generation often results in pattern repetition and poor semantic structure. Multi-Diffusion (Bar-Tal et al., 2023) proposes a sliding window denoising scheme for panoramic image generation. However, this method suffers from severe pattern repetition, as it primarily focuses on the aggregation of local information. Improved variants based on the sliding window denoising scheme include SyncDiffusion (Lee et al., 2023), Demofusion (Du et al., 2024), AccDiffusion (Lin et al., 2025), and CutDiffusion (Lin et al., 2024). Specifically, SyncDiffusion incorporates global information by leveraging the gradient of perceptual loss from the predicted denoised images at each denoising step as guidance. Demofusion employs progressive upscaling, skip residuals, and dilated sampling mechanisms to support higher-resolution image generation. AccDiffusion introduces patch-content-aware prompts, while CutDiffusion adopts a coarse-to-fine strategy to mitigate pattern

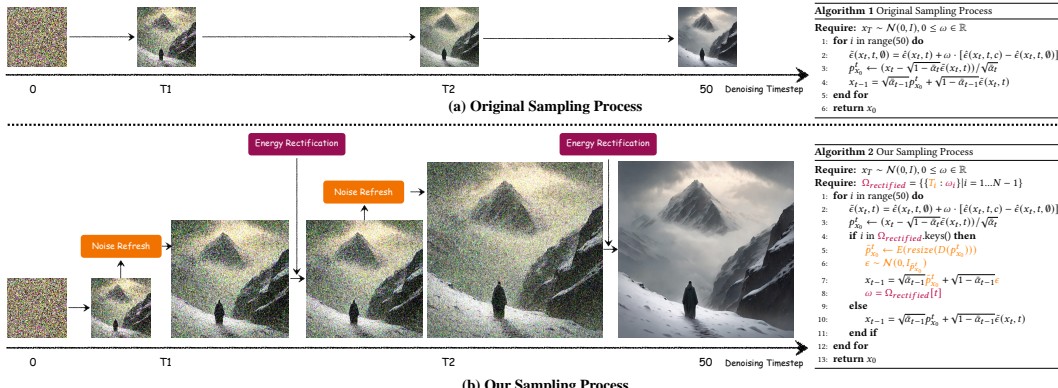

Figure 4: Overview and Pseudo Code of *RectifedHR*. During sampling, we perform Noise Refresh at specific steps, resizing $\tilde{p}_{x_0}^t$ in the RGB space, followed by Energy Rectification, where the classifier-free guidance parameter is appropriately increased to rectify energy decay in the sampling process and thereby recover missing image details.

repetition. Nonetheless, these approaches share complex implementation logic and encounter efficiency bottlenecks due to redundant computation arising from overlapping sliding windows.

ScaleCrafter (He et al., 2023), FouriScale (Huang et al., 2025), HiDiffusion (Zhang et al., 2023b), and Attn-SF (Jin et al., 2023) modify the network architecture of the diffusion model, which may result in suboptimal performance. Furthermore, these methods perform high-resolution denoising throughout the entire sampling process, leading to slower inference compared to our approach, which progressively transitions from low to high resolution. Although HiDiffusion accelerates inference using window attention mechanisms, our method remains faster, as demonstrated by experimental results.

Upscale Guidance (Hwang et al., 2024) and ElasticDiffusion (Haji-Ali et al., 2024) both propose incorporating global and local denoising information into classifier-free guidance (Ho & Salimans, 2022). The global branch of Upscale Guidance and the overlapping window regions in the local branch of ElasticDiffusion involve significantly higher computational complexity compared to our progressive resolution increase strategy. ResMaster (Shi et al., 2024) and HiPrompt (Liu et al., 2024b) introduce multi-modal models to regenerate prompts and enrich image details; however, the use of such multi-modal models introduces substantial overhead, leading to further efficiency issues.

DiffuseHigh (Kim et al., 2024), MegaFusion (Wu et al., 2024), FreCas (Zhang et al., 2024), and AP-LDM (Cao et al., 2024) leverage the detail enhancement capabilities of SDEdit (Meng et al., 2021), progressively adding details from low-resolution to high-resolution images. In contrast to these methods, our approach neither increases sampling steps nor requires additional computations involving classifier-free guidance (CFG) variants, resulting in greater efficiency. Moreover, we identify the issue of energy decay and show that simply adjusting the classifier-free guidance parameter is sufficient to rectify the energy and achieve improved results.

## 3 METHOD

### 3.1 PRELIMINARIES

Diffusion models establish a mapping between Gaussian noise and images, enabling image generation by randomly sampling noise. In this paper, we assume 50 sampling steps, with the denoising process starting at step 0 and ending at step 49. We define $I_o$ as the RGB image. During training, the diffusion model first employs a VAE encoder $E(\cdot)$ to transform the RGB image into a lower-dimensional latent representation, denoted as $x_0$. The forward diffusion process is then defined as:

$$x_t = \sqrt{\bar{\alpha}_t}x_0 + \sqrt{1 - \bar{\alpha}_t}\epsilon. \tag{1}$$

Noise of varying intensity is added to $x_0$ to produce different $x_t$, where $\bar{\alpha}_t$ is a time-dependent scheduler parameter controlling the noise strength, and $\epsilon$ is randomly sampled Gaussian noise. The diffusion model $\hat{\epsilon}(x_t, t, c)$, parameterized by $\theta$, is optimized to predict the added noise via the following training objective:

$$\min_\theta \mathbb{E}_{x_t,t,c} \left[ \| \epsilon - \hat{\epsilon}(x_t, t, c) \|_2^2 \right], \tag{2}$$

where $c$ denotes the conditioning signal for generation (e.g., a text prompt in T2I tasks). During inference, random noise is sampled in the latent space, and the diffusion model gradually transforms this noise into an image via a denoising process. Finally, the latent representation is passed through the decoder $D(\cdot)$ of the VAE to reconstruct the generated RGB image. The objective of high-resolution synthesis is to produce images at resolutions beyond those seen during training—for instance, resolutions exceeding $1024 \times 1024$ in our setting.

**Classifier-free guidance for diffusion models.** Classifier-free guidance (CFG) (Ho & Salimans, 2022) is currently widely adopted to enhance the quality of generated images by incorporating un-conditional outputs at each denoising step. The formulation of classifier-free guidance is as follows:

$$\tilde{\epsilon}(x_t, t) = \hat{\epsilon}(x_t, t, \emptyset) + \omega \cdot [\hat{\epsilon}(x_t, t, c) - \hat{\epsilon}(x_t, t, \emptyset)], \tag{3}$$

where $\omega$ is the hyperparameter of classifier-free guidance, $\hat{\epsilon}(x_t, t, \emptyset)$ and $\hat{\epsilon}(x_t, t, c)$ denote the predicted noises from the unconditional and conditional branches, respectively. We refer to $\tilde{\epsilon}(x_t, t)$ as the predicted noise after applying classifier-free guidance.

**Sampling process for diffusion models.** In this paper, we adopt the DDIM sampler (Song et al., 2020a) as the default. The deterministic sampling formulation of DDIM is given as follows:

$$x_{t-1} = \sqrt{\bar{\alpha}_{t-1}} \underbrace{\left( \frac{x_t - \sqrt{1 - \bar{\alpha}_t} \cdot \tilde{\epsilon}(x_t, t)}{\sqrt{\bar{\alpha}_t}} \right)}_{\text{predicted } x_0 \to p_{x_0}^t} + \sqrt{1 - \bar{\alpha}_{t-1}} \cdot \tilde{\epsilon}(x_t, t). \tag{4}$$

As illustrated in Eq. 4, at timestep $t$, we first predict the noise $\tilde{\epsilon}(x_t, t)$ using the pre-trained neural network $\hat{\epsilon}(\cdot)$. We then compute a "predicted $x_0$" at timestep $t$, denoted as $p_{x_0}^t$. Finally, $x_{t-1}$ is derived from $\tilde{\epsilon}(x_t, t)$ and $p_{x_0}^t$ using the diffusion process defined in Eq. 4.

In this paper, we propose RectifiedHR, which consists of noise refresh and energy rectification. The noise refresh module progressively increases the resolution during the sampling process, while the energy rectification module enhances the visual details of the generated contents.

## 3.2 NOISE REFRESH

To enable high-resolution synthesis, we propose a progressive resizing strategy during sampling. A straightforward baseline for implementing this strategy is to directly perform image-space interpolation in the latent space. However, this approach presents two key issues. First, since the latent space is obtained via VAE compression of the image, interpolation operations that work in RGB space are ineffective in the latent space, as demonstrated by Method D in the ablation study (Table 2). Second, because the latent space consists of $p_{x_0}^t$ and noise, directly resizing it alters the noise distribution, potentially shifting the latent representation outside the diffusion model's valid domain. To address this, we visualize $p_{x_0}^t$, as shown in Fig. 2, and observe that the image corresponding to $p_{x_0}^t$ exhibits RGB-like characteristics in the later stages of sampling. Therefore, we resize $p_{x_0}^t$ to enlarge the latent representation. To ensure the resized latent maintains a Gaussian distribution, we inject new Gaussian noise into $p_{x_0}^t$. The method for enhancing the resolution of $p_{x_0}^t$ is as follows:

$$\tilde{p}_{x_0}^t = E(\text{resize}(D(p_{x_0}^t))), \tag{5}$$

where $E$ denotes the VAE encoder, $D$ denotes the VAE decoder, and $\text{resize}(\cdot)$ refers to the operation of enlarging the RGB image. We adopt bilinear interpolation as the default resizing method. The procedure for re-adding noise is as follows:

$$x_{t-1} = \sqrt{\bar{\alpha}_{t-1}} \tilde{p}_{x_0}^t + \sqrt{1 - \bar{\alpha}_{t-1}} \epsilon, \tag{6}$$

where $\epsilon$ denotes a random Gaussian noise that shares the same shape as $\tilde{p}_{x_0}^t$. We refer to this process as **Noise Refresh**.

As illustrated in Fig. 4b, the noise refresh operation is applied at specific time points $T_i$ during the sampling process. To automate the selection of these timesteps $T$, we propose the following selection formula:

$$T_i = \lfloor (T_{\max} - T_{\min}) * (\frac{i-1}{N})^{M_T} + T_{\min} \rfloor, \tag{7}$$

where $T_{\max}$ and $T_{\min}$ define the range of sampling timesteps at which noise refresh is applied. $N$ denotes the number of different resolutions in the denoising process, and $N - 1$ corresponds to the number of noise refresh operations performed. $N$ is a positive integer, and the range of $i$ includes all integers in $[1, N)$. Specifically, we set $T_0$ to 0 and $T_{\max}$ to the total number of sampling steps. $T_{\min}$ is treated as a hyperparameter. Since $p_{x_0}^t$ exhibits more prominent image features in the later stages of sampling, as shown in Fig. 2, $T_{\min}$ is selected to fall within the later stage of the sampling process. A quantitative analysis of the variation in $p_{x_0}^t$ is provided in Supp. A.5.

## 3.3 ENERGY RECTIFICATION

Although noise refresh enables the diffusion model to generate high-resolution images, we observe that introducing noise refresh during the sampling process leads to blurriness in the generated content, as illustrated in the fourth row of Fig. 8. To investigate the cause of this phenomenon, we introduce the average latent energy formula as follows:

$$\mathbb{E}[x_t^2] = \frac{\sum_{i=1}^C \sum_{j=1}^H \sum_{k=1}^W x_{t_{ijk}}^2}{C \times H \times W}, \tag{8}$$

where $x_t$ represents the latent variable at time $t$, and $C$, $H$, and $W$ denote the channel, height, and width dimensions of the latent, respectively. This definition closely resembles that of image energy and is used to quantify the average energy per element of the latent vector. To investigate the issue of image blurring, we conduct an average latent energy analysis on 100 random prompts. As illustrated in Fig. 3a, we first compare the average latent energy between the noise refresh sampling process and the original sampling process. We observe significant energy decay during the noise refresh sampling process, which explains why the naive implementation produces noticeably blurred images. Subsequently, we experimentally discover that the hyperparameter $\omega$ in classifier-free guidance influences the average latent energy. As shown in Fig. 3b, we find that increasing the classifier-free guidance parameter $\omega$ leads to a gradual increase in energy. Therefore, the issue of energy decay—and thus image quality degradation—can be mitigated by increasing $\omega$ to boost the energy in the noise refresh sampling scheme. As demonstrated in the left image of Fig. 3a, once energy is rectified by using a larger classifier-free guidance hyperparameter $\omega$, the blurriness is substantially reduced, and the generated image exhibits significantly improved clarity. We refer to this process of correcting energy decay as **Energy Rectification**. However, we note that a larger $\omega$ is not always beneficial, as excessively high values may lead to overexposure. The goal of energy rectification is to align the energy level with that of the original diffusion model's denoising process, rather than to maximize energy indiscriminately. The experiment analyzing the rectified average latent energy curve is provided in Supp. A.10.

As shown in Fig. 4b, the energy rectification operation is applied during the sampling process following noise refresh. To automatically select an appropriate $\omega$ value for classifier-free guidance, we propose the following selection formula:

$$\omega_i = (\omega_{\max} - \omega_{\min}) * (\frac{i}{N-1})^{M_\omega} + \omega_{\min}, \tag{9}$$

where $\omega_{\max}$ and $\omega_{\min}$ define the range of $\omega$ values used in classifier-free guidance during the sampling process. $N$ denotes the number of different resolutions in the denoising process, and $N-1$ corresponds to the number of noise refresh operations performed. $N$ is a positive integer, and the range of $i$ includes all integers in $[0, N)$. $\omega_{\min}$ refers to the CFG hyperparameter at the original resolution supported by the diffusion model. $M_\omega$ is a tunable hyperparameter that allows for different strategies in selecting $\omega_i$. The value of $N$ used in Eq. 7 and Eq. 9 remains consistent throughout the sampling process.

Additionally, we establish the connection between energy rectification and SNR correction strategies proposed in (Zhang et al., 2024; Wu et al., 2024; Hoogeboom et al., 2023), showing that SNR correction is essentially a form of energy rectification. The proof is provided in Supp. A.6.

| | Methods | FID$_r$ ↓ | KID$_r$ ↓ | IS$_r$ ↑ | FID$_c$ ↓ | KID$_c$ ↓ | IS$_c$ ↑ | CLIP↑ | Time↓ | User Study↑ |
|---|---|---|---|---|---|---|---|---|---|---|
| 2048 × 2048 | FouriScale | 71.344 | 0.010 | 15.957 | 53.990 | 0.014 | 20.625 | 31.157 | 59s | 11.6% |
| | ScaleCrafter | 64.236 | 0.007 | 15.952 | 45.861 | 0.010 | 22.252 | 31.803 | 35s | 13.6% |
| | HiDiffusion | 63.674 | 0.007 | 16.876 | 41.930 | 0.008 | 23.165 | 31.711 | 18s | 12.7% |
| | CutDiffusion | 59.152 | 0.007 | 17.109 | 38.004 | 0.008 | 23.444 | 32.573 | 53s | - |
| | ElasticDiffusion | 56.639 | 0.010 | 15.326 | 37.649 | 0.014 | 19.867 | 32.301 | 150s | - |
| | AccDiffusion | 48.143 | **0.002** | 18.466 | 32.747 | 0.008 | 24.778 | 33.153 | 111s | 13.8% |
| | DiffuseHigh | 49.748 | 0.003 | 19.537 | 27.667 | 0.004 | 27.876 | 33.436 | 37s | - |
| | FreCas | 49.129 | 0.003 | 20.274 | 27.002 | 0.004 | **29.843** | 33.700 | 14s | 16.2% |
| | DemoFusion | **47.079** | **0.002** | 19.533 | 26.441 | 0.004 | 27.843 | 33.748 | 79s | - |
| | Ours | 48.361 | **0.002** | 20.616 | 25.347 | **0.003** | 28.126 | **33.756** | **13s** | **32.2%** |
| 4096 × 4096 | FouriScale | 135.111 | 0.046 | 9.481 | 129.895 | 0.057 | 9.792 | 26.891 | 489s | 11.6% |
| | ScaleCrafter | 110.094 | 0.028 | 10.098 | 112.105 | 0.043 | 11.421 | 27.809 | 528s | 13.6% |
| | HiDiffusion | 93.515 | 0.024 | 11.878 | 120.170 | 0.058 | 11.272 | 27.853 | 71s | 12.7% |
| | CutDiffusion | 130.207 | 0.055 | 9.334 | 113.033 | 0.055 | 10.961 | 26.734 | 193s | - |
| | ElasticDiffusion | 101.313 | 0.056 | 9.406 | 111.102 | 0.089 | 7.627 | 27.725 | 400s | - |
| | AccDiffusion | 54.918 | 0.005 | 17.444 | 60.362 | 0.023 | 16.370 | 32.438 | 826s | 13.8% |
| | DiffuseHigh | 48.861 | **0.003** | 19.716 | 40.267 | 0.010 | 21.550 | 33.390 | 190s | - |
| | FreCas | 49.764 | **0.003** | 18.656 | 39.047 | 0.010 | **21.700** | 33.237 | 74s | 16.2% |
| | DemoFusion | 48.983 | **0.003** | 18.225 | 38.136 | 0.010 | 20.786 | 33.311 | 605s | - |
| | Ours | **48.684** | **0.003** | 20.352 | **35.718** | **0.009** | 20.819 | **33.415** | 37s | **32.2%** |

Table 1: Comparison to SOTA methods at 2048 × 2048 and 4096 × 4096 resolutions. Bold numbers indicate the best performance, while underlined numbers denote the second-best performance.

## 4 EXPERIMENTS

### 4.1 EVALUATION SETUP

Our experiments use SDXL (Podell et al., 2023) as the base model, which by default generates images at a resolution of 1024 × 1024. Furthermore, our method can also be applied to Stable Diffusion and transformer-based diffusion models such as WAN (Wang et al., 2025) and SD3 (Esser et al., 2024), as demonstrated in Fig. 5 and Supp. A.7. The specific evaluation metrics and methods are provided in Supp. A.11. The comparison includes state-of-the-art training-free methods: Demofusion (Du et al., 2024), DiffuseHigh (Kim et al., 2024), HiDiffusion (Zhang et al., 2023b), CutDiffusion (Lin et al., 2024), ElasticDiffusion (Haji-Ali et al., 2024), FreCas (Zhang et al., 2024), FouriScale (Huang et al., 2025), ScaleCrafter (He et al., 2023), and AccDiffusion (Lin et al., 2025). Quantitative assessments focus on upsampling to 2048 × 2048 and 4096 × 4096 resolutions. All baseline methods are fairly and fully reproduced. For the 2048 × 2048 resolution setting, we set $T_{\min}$ to 40, $T_{\max}$ to 50, $N$ to 2, $\omega_{\min}$ to 5, $\omega_{\max}$ to 30, $M_T$ to 1, and $M_\omega$ to 1. For the 4096 × 4096 resolution setting, we set $T_{\min}$ to 40, $T_{\max}$ to 50, $N$ to 3, $\omega_{\min}$ to 5, $\omega_{\max}$ to 50, $M_T$ to 0.5, and $M_\omega$ to 0.5. All experiments are conducted using 8 NVIDIA A800 GPUs unless specified. The above hyperparameters are obtained through a hyperparameter search, with detailed ablation studies provided in Supp. A.8. More qualitative results are presented in Supp. A.2 and Supp. A.13.

### 4.2 QUANTITATIVE RESULTS

As shown in Tab. 1, *RectifiedHR* consistently surpasses competing methods at both 2048 × 2048 and 4096 × 4096. At 2048 × 2048, it leads 6/8 metrics, placing second on one and third on another; at 4096 × 4096, it leads 7/8 and places third on the remaining metric. At 2048 × 2048, our KID$_r$ ranks third because this metric downsamples high-resolution images for evaluation, underrepresenting fine details—a known limitation (Du et al., 2024; Lin et al., 2025). Although *RectifiedHR* ranks second and third on IS$_c$, its dominance on the other metrics, together with strong computational efficiency, demonstrates its overall effectiveness and robustness for high-resolution generation. When scaled to 4096 × 4096, *RectifiedHR* is roughly twice as fast as the next fastest approach. This speedup comes from preserving the original number of sampling steps and carefully tuning the CFG hyperparameter. In contrast, methods such as DiffuseHigh incur substantial overhead by adding extra sampling via repeated SDEdit and FreCas within heavier CFG pipelines. Notably, *RectifiedHR* achieves this speed without sacrificing quality, matching or exceeding baseline visual fidelity across resolutions, thereby striking a favorable speed–quality balance. User study also demonstrates the advantages of our approach. Details of the user study are presented in Supp. A.14. Since the images of all resolutions were mixed together during the user study, the user study values in different resolutions are the same.

| | Methods | Noise Refresh | Energy Rectification | Resize Latent | $FID_r \downarrow$ | $KID_r \downarrow$ | $IS_r \uparrow$ | $FID_c \downarrow$ | $KID_c \downarrow$ | $IS_c \uparrow$ | CLIP $\uparrow$ |
|---|---|---|---|---|---|---|---|---|---|---|---|
| 2048×2048 | A | × | × | × | 98.676 | 0.030 | 13.193 | 73.426 | 0.029 | 17.867 | 30.021 |
| | B | × | ✓ | × | 86.595 | 0.021 | 13.900 | 60.625 | 0.021 | 19.921 | 30.728 |
| | C | ✓ | × | × | 79.743 | 0.021 | 13.334 | 76.023 | 0.035 | 11.840 | 29.966 |
| | D | × | ✓ | ✓ | 78.307 | 0.019 | 13.221 | 74.419 | 0.034 | 11.883 | 29.523 |
| | Ours | ✓ | ✓ | × | **48.361** | **0.002** | **20.616** | **25.347** | **0.003** | **28.126** | **33.756** |
| 4096×4096 | A | × | × | × | 187.667 | 0.088 | 8.636 | 111.117 | 0.057 | 13.383 | 25.447 |
| | B | × | ✓ | × | 175.830 | 0.079 | 8.403 | 80.733 | 0.034 | 15.791 | 26.099 |
| | C | ✓ | × | × | 85.088 | 0.026 | 13.114 | 141.422 | 0.091 | 5.465 | 29.548 |
| | D | × | ✓ | ✓ | 89.968 | 0.033 | 11.973 | 145.472 | 0.103 | 6.312 | 28.212 |
| | Ours | ✓ | ✓ | × | **48.684** | **0.003** | **20.352** | **35.718** | **0.009** | **20.819** | **33.415** |

Table 2: Quantitative results of the ablation studies. Method A denotes direct inference (without noise refresh and energy rectification), Method B excludes noise refresh, Method C excludes energy rectification, and Method D replaces noise refresh in our method with direct latent resizing. Ours refers to the full version of our proposed method.

## 4.3 ABLATION STUDY

To evaluate the effectiveness of each module in our method, we conduct both quantitative experiments (Tab. 2) and qualitative experiments (Supp. A.4). The metric computation follows the procedure described in Supp. A.11. All hyperparameters are set according to Sec. 4.1. Additionally, in scenarios without energy rectification, the classifier-free guidance hyperparameter $\omega$ is fixed at 5. For simplicity, this section mainly compares the $FID_c$ metric at the $4096 \times 4096$ resolution. Comparing Method B in Tab.2 with Ours, the $FID_c$ increases from 35.718 to 80.733 without noise refresh. Comparing Method C in Tab. 2 with Ours, the $FID_c$ rises sharply from 35.718 to 141.422 without energy rectification, demonstrating that energy decay severely degrades generation quality. This underscores the importance of energy rectification—despite its simplicity, it yields significant improvements. Comparing Method D in Tab. 2 with Ours, the $FID_c$ improves from 145.472 to 35.718, revealing that directly resizing the latent is ineffective. This confirms that noise refresh is indispensable and cannot be replaced by naïve latent resizing. We also conduct ablation studies on the hyperparameters related to Eq. 7 and Eq. 9, with detailed results provided in Supp. A.8.

## 5 MORE APPLICATIONS

This section highlights how *RectifiedHR* can enhance a variety of tasks, with a focus on demonstrating visual improvements. The experiments cover diverse tasks, models, and sampling methods to validate the effectiveness of our approach. While primarily evaluated on classic methods and models, *RectifiedHR* can also be seamlessly integrated into more advanced techniques. Supp. A.9 provides detailed quantitative results and corresponding hyperparameter settings.

**Video Generation.** *RectifiedHR* can be directly applied to video diffusion models such as WAN (Wang et al., 2025). The officially supported maximum resolution for WAN 1.3B is $480 \times 832$. As shown in Fig. 5a and Tab. 3, directly generating high-resolution videos with WAN may lead to generation failure or prompt misalignment. However, integrating *RectifiedHR* enables WAN to produce high-quality, high-resolution videos reliably. More experimental results and details are presented in Supp. A.12 and Supp. A.9.

| | Visual Quality $\uparrow$ | Motion Quality $\uparrow$ | Temporal Consistency $\uparrow$ |
|---|---|---|---|
| Direct Inference | 65.31 | 51.91 | 63.78 |
| Ours | **67.22** | **54.30** | **64.26** |

Table 3: Quantitative results of video generation.

**Image Editing.** *RectifiedHR* can be applied to image editing tasks. In this section, we use SDXL as the base model with a default resolution of $1024 \times 1024$. Directly editing high-resolution images with OIR often results in ghosting artifacts, as illustrated in rows a, b, d, and e of Fig. 5b. Additionally, it can cause shape distortions and deformations, as shown in rows c and f. In contrast, the combination of OIR and *RectifiedHR* effectively mitigates these issues, as demonstrated in Fig. 5b.

**Customized Generation.** *RectifiedHR* can be directly adapted to DreamBooth using SD1.4 with a default resolution of $512 \times 512$, as shown in Fig. 5c. The direct generation of high-resolution cus-

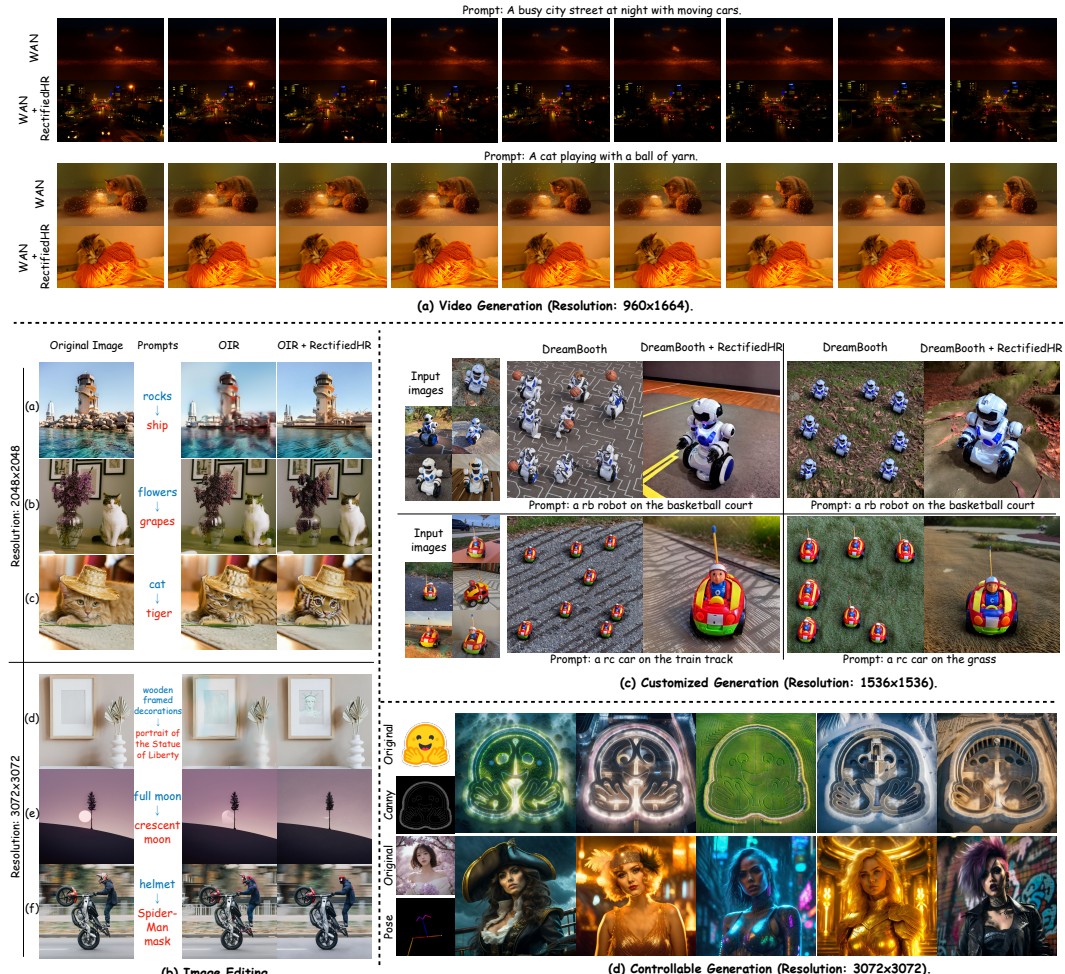

Figure 5: Applications. (a) Video Generation. (b) Image Editing. (c) Customized Generation. (d) Controllable Generation. **Contents are best viewed when zoomed in.**

tomized images often leads to severe repetitive pattern artifacts. Integrating *RectifiedHR* effectively addresses this problem.

**Controllable Generation.** *RectifiedHR* can be seamlessly integrated with ControlNet (Zhang et al., 2023a) using SDXL at a default resolution of $1024 \times 1024$ to enable controllable generation. As shown in Fig. 5d, control signals may include pose, canny edges, and other modalities.

# 6 CONCLUSION AND FUTURE WORK

We propose an efficient and straightforward method, *RectifiedHR*, for high-resolution synthesis. Specifically, we conduct an average latent energy analysis and, to the best of our knowledge, are the first to identify the energy decay phenomenon during high-resolution synthesis. Our approach introduces a novel training-free pipeline that is both simple and effective, primarily incorporating noise refresh and energy rectification operations. Extensive comparisons demonstrate that *RectifiedHR* outperforms existing methods in both effectiveness and efficiency. Nonetheless, our method has certain limitations. During the noise refresh stage, it requires both decoding and encoding operations via the VAE, which impacts the overall runtime. In future work, we aim to investigate performing resizing operations directly in the latent space to further improve efficiency.

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

## A    SUPPLEMENTARY

### A.1    USE OF LLMS

We use LLMs to polish my papers, correct some grammatical errors, and make the language more concise and fluent.

## A.2 QUALITATIVE RESULTS

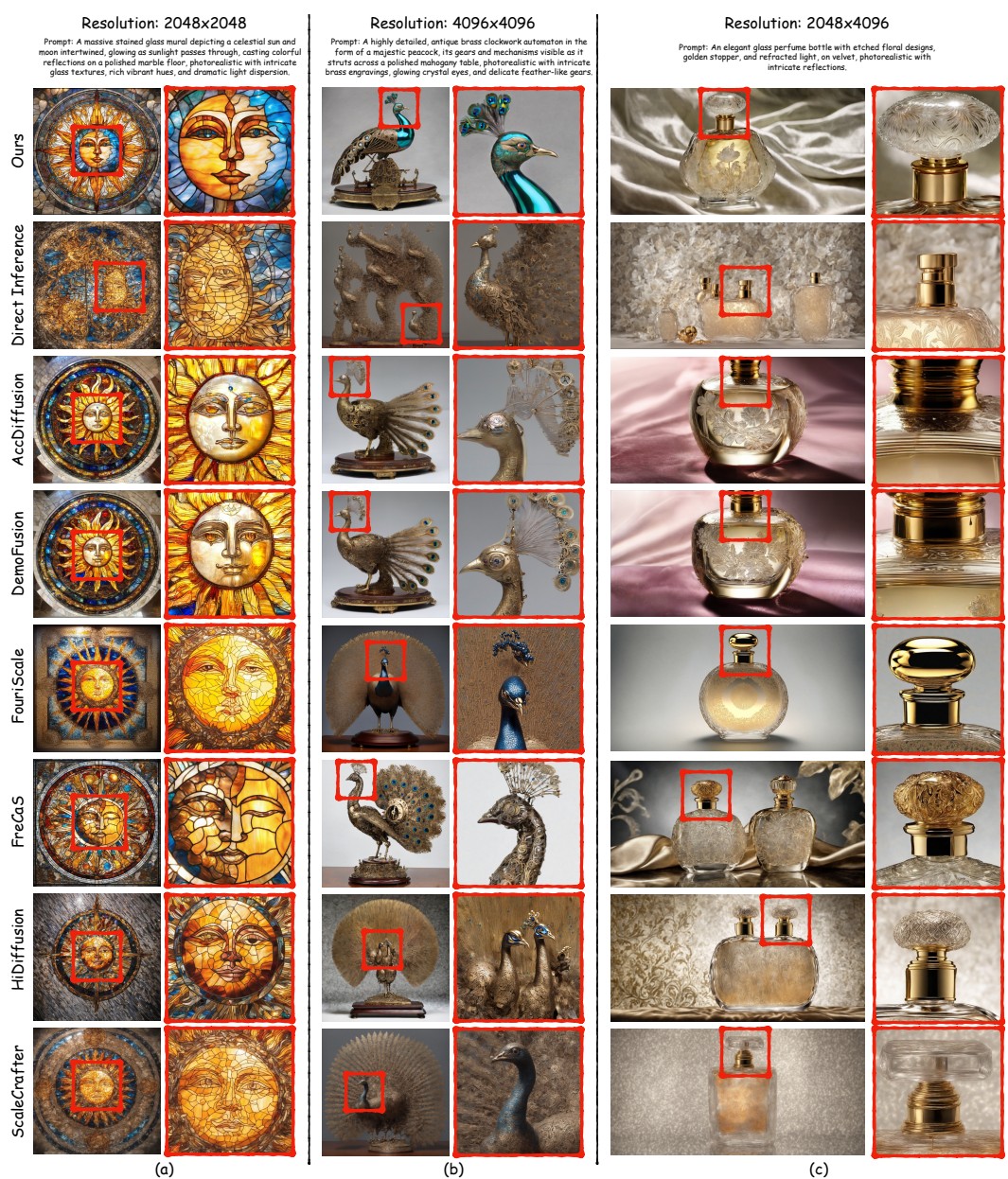

Figure 6: Qualitative comparison across three different resolutions between our method and other training-free methods. The red box indicates an enlarged view of a local region within the high-resolution image.

As shown in Fig. 6, to clearly illustrate the differences between our method and existing baselines, we select a representative prompt for each of the three resolution scenarios and conduct qualitative comparisons against SDXL direct inference, AccDiffusion, DemoFusion, FouriScale, FreCas, HiDiffusion, and ScaleCrafter. AccDiffusion and DemoFusion tend to produce blurry details and lower visual quality, such as the peacock's eyes and feathers in column b, and the bottle stoppers in column c. FouriScale and ScaleCrafter often generate deformed or blurred objects that fail to satisfy the prompt, such as feathers lacking peacock characteristics in column b, and a blurry bottle body missing the velvet element specified in the prompt in column c. HiDiffusion may introduce repetitive patterns, as seen in the duplicate heads in column b and the recurring motifs on the bottles

in column c. FreCas can produce distorted details or fail to adhere to the prompt, such as the deformed and incorrect number of bottles in column c. In contrast, our method consistently achieves superior visual quality across all resolutions. In column a, our approach generates the clearest and most refined faces and is the only method that correctly captures the prompt's description of the sun and moon intertwined. In column b, our peacock is the most detailed and visually accurate, with a color distribution and fine-grained features that closely align with the prompt's reference to crystal eyes and delicate feather-like gears. In column c, our method demonstrates the highest fidelity in rendering the bottle stopper and floral patterns, and it uniquely preserves the white velvet background described in the prompt. These qualitative results highlight the effectiveness of our method in generating visually consistent, detailed, and prompt-faithful images across different resolution settings.

### A.3 COMPARISON WITH THE SUPER-RESOLUTION MODEL

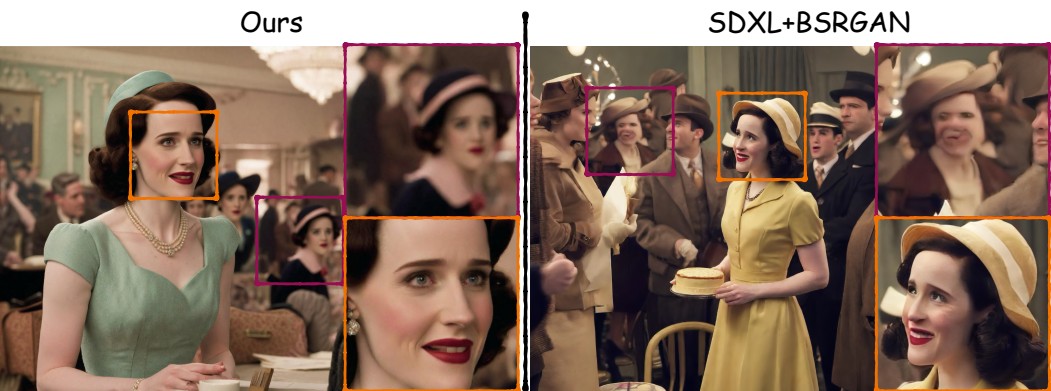

Figure 7: Qualitative comparison between our method and SDXL+BSRGAN at a resolution of $2048 \times 2048$.

Training-free high-resolution image generation methods primarily exploit intrinsic properties of diffusion models to achieve super-resolution. Beyond the aforementioned approaches, another viable strategy adopts a two-stage pipeline that combines diffusion models with dedicated super-resolution models. For example, methods such as SDXL + BSRGAN first generate an image using a diffusion model, then apply a super-resolution model to upscale it to the target resolution. To further evaluate the differences between SDXL+BSRGAN and our method, we conduct additional qualitative comparisons. The experimental setup follows that described in Sec. 4.1. As shown in Fig. 7, we observe that when images generated by SDXL exceed the domain of the original training data—such as in cases involving distorted facial features—BSRGAN is unable to correct these artifacts, resulting in performance degradation. Furthermore, existing two-stage approaches rely on pre-trained super-resolution models constrained by fixed-resolution training data. In contrast, our method inherently adapts to arbitrary resolutions without retraining. For example, as demonstrated in the $2048 \times 4096$ resolution scene in Fig. 6, our approach remains effective, whereas BSRGAN cannot be applied.

## A.4 QUALITATIVE ABLATION STUDY

Figure 8: Qualitative results of the ablation studies at $2048 \times 2048$ resolution. The orange and blue boxes indicate enlarged views of local regions within the high-resolution image. Zoom in for details.

To evaluate the effectiveness of each module in our method, we conduct qualitative experiments (Fig. 8). All hyperparameters are set according to Sec. 4.1. Additionally, in scenarios without energy rectification, the classifier-free guidance hyperparameter $\omega$ is fixed at 5. As shown in Fig. 8c vs. Fig. 8e, this performance drop is due to the failure in generating correct semantic structures caused by the absence of noise refresh. Fig. 8d and Fig. 8e highlight the critical role of energy rectification in enhancing fine details. This confirms that noise refresh is indispensable and cannot be replaced by naïve latent resizing.

## A.5 QUANTITATIVE ANALYSIS OF "PREDICTED $x_0$"

To quantitatively validate this observation, as shown in Fig.9, we conduct additional experiments on the generation of $p_{x_0}^t$ using 100 random prompts sampled from LAION-5B (Schuhmann et al., 2022), and analyze the CLIP Score (Hessel et al., 2021) and Mean Squared Error (MSE). From Fig. 9a, we observe that after 30 denoising steps, the MSE between $p_{x_0}^t$ and $p_{x_0}^{t-1}$ exhibits minimal change. In Fig. 9b, we find that the CLIP score between $p_{x_0}^t$ and the corresponding prompt increases slowly beyond 30 denoising steps.

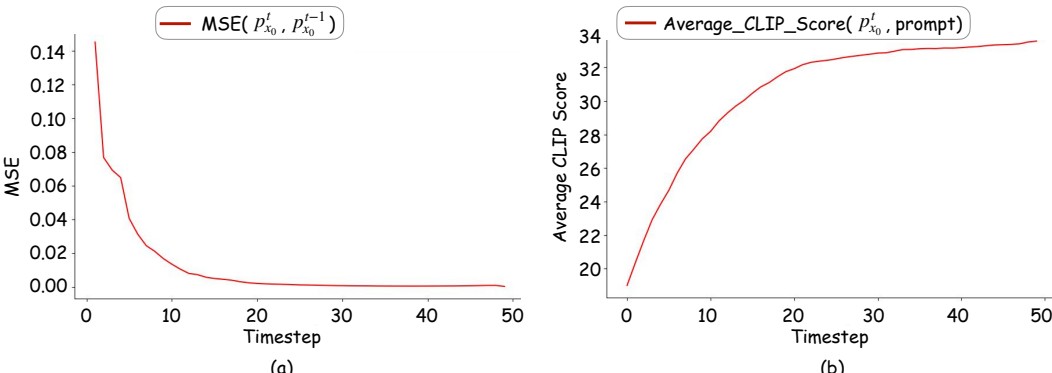

Figure 9: The trend of the "predicted $x_0$" at different timesteps $t$, denoted as $p_{x_0}^t$, evaluated on 100 random prompts. (a) The average MSE between $p_{x_0}^t$ and $p_{x_0}^{t-1}$. The x-axis represents the sampling timestep, and the y-axis denotes the average MSE. It can be observed that after approximately 30 steps, the rate of change in $p_{x_0}^t$ slows significantly. (b) The trend of the average CLIP Score between $p_{x_0}^t$ and the prompt across different timesteps. The x-axis represents the sampling timestep, and the y-axis denotes the average CLIP Score.

### A.6 THE CONNECTION BETWEEN ENERGY RECTIFICATION AND SIGNAL-TO-NOISE RATIO (SNR) CORRECTION

In the proof presented in this section, all symbols follow the definitions provided in the Method section of the main text. Any additional symbols not previously defined will be explicitly specified. This proof analyzes energy variation using the DDIM sampler as an example. The sampling formulation of DDIM is given as follows:

$$
\begin{aligned}
x_{t-1} &= \sqrt{\bar{\alpha}_{t-1}}\left(\frac{x_t - \sqrt{1-\bar{\alpha}_t}\tilde{\epsilon}(x_t,t)}{\sqrt{\bar{\alpha}_t}}\right) + \sqrt{1-\bar{\alpha}_{t-1}} \cdot \tilde{\epsilon}(x_t,t) \\
&= \sqrt{\frac{\bar{\alpha}_{t-1}}{\bar{\alpha}_t}}x_t + \left(\sqrt{1-\bar{\alpha}_{t-1}} - \frac{\sqrt{\bar{\alpha}_{t-1}}\sqrt{1-\bar{\alpha}_t}}{\sqrt{\bar{\alpha}_t}}\right)\tilde{\varepsilon}(x_t,t).
\end{aligned}
\tag{10}
$$

To simplify the derivation, we assume that all quantities in the equation are scalar values. Based on the definition of average latent energy in Eq.8 of the main text, the average latent energy during the DDIM sampling process can be expressed as follows:

$$
\begin{aligned}
\mathbb{E}[x_{t-1}^2] &= \mathbb{E}\left[\sqrt{\frac{\bar{\alpha}_{t-1}}{\bar{\alpha}_t}}x_t\right]^2 + \mathbb{E}\left[\left(\sqrt{1-\bar{\alpha}_{t-1}} - \frac{\sqrt{\bar{\alpha}_{t-1}}\sqrt{1-\bar{\alpha}_t}}{\sqrt{\bar{\alpha}_t}}\right)\tilde{\varepsilon}(x_t,t)\right]^2 \\
&+ 2 \times \mathbb{E}\left[\sqrt{\frac{\bar{\alpha}_{t-1}}{\bar{\alpha}_t}}x_t\right] \times \mathbb{E}\left[\left(\sqrt{1-\bar{\alpha}_{t-1}} - \frac{\sqrt{\bar{\alpha}_{t-1}}\sqrt{1-\bar{\alpha}_t}}{\sqrt{\bar{\alpha}_t}}\right)\tilde{\varepsilon}(x_t,t)\right].
\end{aligned}
\tag{11}
$$

We assume that the predicted noise $\tilde{\epsilon}$ follows a standard normal distribution, such that $\mathbb{E}[\tilde{\epsilon}(x_t,t)] = 0$. Under this assumption, the average latent energy of the DDIM sampler can be simplified as:

$$
\mathbb{E}[x_{t-1}^2] = \mathbb{E}\left[\sqrt{\frac{\bar{\alpha}_{t-1}}{\bar{\alpha}_t}}x_t\right]^2 + \mathbb{E}\left[\left(\sqrt{1-\bar{\alpha}_{t-1}} - \frac{\sqrt{\bar{\alpha}_{t-1}}\sqrt{1-\bar{\alpha}_t}}{\sqrt{\bar{\alpha}_t}}\right)\tilde{\varepsilon}(x_t,t)\right]^2.
\tag{12}
$$

Several previous works (Hoogeboom et al., 2023; Zhang et al., 2024; Wu et al., 2024; Hwang et al., 2024) define the Signal-to-Noise Ratio (SNR) at a given timestep of a diffusion model as follows:

$$
SNR_t = \frac{\bar{\alpha}_t}{1-\bar{\alpha}_t}.
\tag{13}
$$

Several works (Hoogeboom et al., 2023; Zhang et al., 2024; Wu et al., 2024; Hwang et al., 2024) have observed that the SNR must be adjusted during the generation process at different resolutions.

Suppose the diffusion model is originally designed for a resolution of $H \times W$, and we aim to extend it to generate images at a higher resolution of $H' \times W'$, where $H' > H$ and $W' > W$. According to the derivations in (Zhang et al., 2024; Wu et al., 2024), the adjusted formulation of $\alpha_t$ is given as follows:

$$\bar{\alpha}_t' = \frac{\bar{\alpha}_t}{\gamma - (\gamma - 1)\bar{\alpha}_t}. \tag{14}$$

Here, the value of $\gamma$ is typically defined as $(H'/H \cdot W'/W)^2$. By substituting the modified $\bar{\alpha}_t'$ into Eq. 10, we obtain the SNR-corrected sampling formulation as follows:

$$\mathbb{E}[x_{t-1}] = \sqrt{\frac{\bar{\alpha}_{t-1}'}{\bar{\alpha}_t'}}\mathbb{E}[x_t] + \left(\sqrt{1 - \bar{\alpha}_{t-1}'} - \frac{\sqrt{\bar{\alpha}_{t-1}'}\sqrt{1 - \bar{\alpha}_t'}}{\sqrt{\bar{\alpha}_t'}}\right)\mathbb{E}[\tilde{\epsilon}(x_t, t)]$$

$$= \sqrt{\frac{\frac{\bar{\alpha}_{t-1}}{\gamma - (\gamma-1)\bar{\alpha}_{t-1}}}{\frac{\bar{\alpha}_t}{\gamma - (\gamma-1)\bar{\alpha}_t}}}\mathbb{E}[x_t] + \left(\sqrt{1 - \frac{\bar{\alpha}_{t-1}}{\gamma - (\gamma-1)\bar{\alpha}_{t-1}}} - \sqrt{\frac{\frac{\bar{\alpha}_{t-1}}{\gamma - (\gamma-1)\bar{\alpha}_{t-1}}\left(1 - \frac{\bar{\alpha}_t}{\gamma - (\gamma-1)\bar{\alpha}_t}\right)}{\frac{\bar{\alpha}_t}{\gamma - (\gamma-1)\bar{\alpha}_t}}}\right)\mathbb{E}[\tilde{\epsilon}(x_t, t)] \tag{15}$$

$$= \sqrt{\frac{\gamma - (\gamma-1)\bar{\alpha}_t}{\gamma - (\gamma-1)\bar{\alpha}_{t-1}}}\sqrt{\frac{\bar{\alpha}_{t-1}}{\bar{\alpha}_t}}\mathbb{E}[x_t] + \sqrt{\frac{\gamma}{\gamma - (\gamma-1)\bar{\alpha}_{t-1}}}\left(\sqrt{1 - \bar{\alpha}_{t-1}} - \frac{\sqrt{\bar{\alpha}_{t-1}}\sqrt{1 - \bar{\alpha}_t}}{\sqrt{\bar{\alpha}_t}}\right)\mathbb{E}[\tilde{\varepsilon}(x_t, t)].$$

The average latent energy under SNR correction can be derived as follows:

$$\mathbb{E}[x_{t-1}^2] = \mathbb{E}\left[\sqrt{\frac{\bar{\alpha}_{t-1}'}{\bar{\alpha}_t'}}x_t\right]^2 + \mathbb{E}\left[\left(\sqrt{1 - \bar{\alpha}_{t-1}'} - \frac{\sqrt{\bar{\alpha}_{t-1}'}\sqrt{1 - \bar{\alpha}_t'}}{\sqrt{\bar{\alpha}_t'}}\right)\tilde{\epsilon}(x_t, t)\right]^2$$

$$= \frac{\gamma - (\gamma-1)\bar{\alpha}_t}{\gamma - (\gamma-1)\bar{\alpha}_{t-1}}\mathbb{E}\left[\sqrt{\frac{\bar{\alpha}_{t-1}}{\bar{\alpha}_t}}x_t\right]^2 + \frac{\gamma}{\gamma - (\gamma-1)\bar{\alpha}_{t-1}}\mathbb{E}\left[\left(\sqrt{1 - \bar{\alpha}_{t-1}} - \frac{\sqrt{\bar{\alpha}_{t-1}}\sqrt{1 - \bar{\alpha}_t}}{\sqrt{\bar{\alpha}_t}}\right)\tilde{\epsilon}(x_t, t)\right]^2. \tag{16}$$

Compared to the original energy formulation Eqa. 12, two additional coefficients appear: $\frac{\gamma - (\gamma-1)\bar{\alpha}_t}{\gamma - (\gamma-1)\bar{\alpha}_{t-1}}$ and $\frac{\gamma}{\gamma - (\gamma-1)\bar{\alpha}_{t-1}}$. Since $\bar{\alpha}_{t-1}$ and $\bar{\alpha}_t$ are very close, the first coefficient is approximately equal to 1. In the DDIM sampling formulation, $\bar{\alpha}_t$ is within the range $[0, 1]$, which implies that the second coefficient falls within $[1, \gamma]$. As a result, after the SNR correction, the average latent energy increases. Therefore, SNR correction essentially serves as a mechanism for energy enhancement. In this sense, both energy rectification and SNR correction aim to increase the average latent energy. However, since our method allows for the flexible selection of hyperparameters, it can achieve superior performance.

## A.7 APPLYING *RectifiedHR* TO STABLE DIFFUSION 3

To validate the effectiveness of our method on a transformer-based diffusion model, we apply it to `stable-diffusion-3-medium` using the `diffusers` library. As shown in Fig. 10, we compare the qualitative results of our method with those of direct inference at a resolution of $2048 \times 2048$. It can be observed that direct inference introduces grid artifacts and object deformations, whereas our method partially mitigates and corrects these issues.

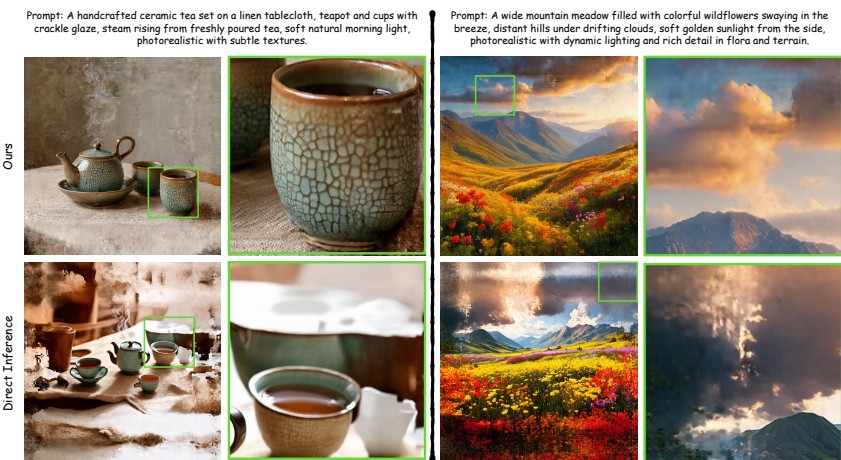

Figure 10: Qualitative comparison on Stable Diffusion 3 at $2048 \times 2048$ resolution. The green boxes indicate enlarged views of local regions within the high-resolution image.

In addition, as shown in Tab. 4, we provide additional quantitative results on SD3 (50 images, 2048×2048), and the test results mainly include CLIP-Score (Hessel et al., 2021) and DEQA-Score (You et al., 2025).

| Model:SD3 | CLIP-Score↑ | DEQA-score↑ |
|---|---|---|
| Direct-Inference | 0.275 | 3.311 |
| RectifiedHR | **0.289** | **3.621** |

Table 4: The quantitative results of SD3.

## A.8 ABLATION RESULTS ON HYPERPARAMETERS

In this section, we conduct ablation experiments on the hyperparameters in Eq.7 and Eq.9 of the main text using SDXL. The baseline hyperparameter settings follow those described in the Evaluation Setup section of the main text. We vary one hyperparameter at a time while keeping the others fixed at the two target resolutions to evaluate the impact of each parameter on performance, as defined in Eq.7 and Eq.9 of the main text. The evaluation procedure for $\text{FID}_c$, $\text{FID}_r$, $\text{IS}_c$, and $\text{IS}_r$ follows the protocol outlined in Sec. A.11. All experiments are conducted on two NVIDIA A800 GPUs unless otherwise specified. As a result, the performance may differ slightly from experiments conducted using eight NVIDIA A800 GPUs.

In Eq.7 and Eq.9 of the main text, $\omega_{\min}$ and $T_{\max}$ are fixed and do not require ablation. The value of $N$ in both equations is kept consistent. For the $2048 \times 2048$ resolution scene, with $N$ set to 2, variations in $M_T$ and $M_\omega$ do not significantly affect the results. Thus, only $N$, $\omega_{\max}$, and $T_{\min}$ are ablated. The quantitative ablation results for the $2048 \times 2048$ resolution are shown in Fig. 11, Fig. 12, and Fig. 13. For the $4096 \times 4096$ resolution scene, $N$, $\omega_{\max}$, $T_{\min}$, $M_T$, and $M_\omega$ are ablated. The corresponding quantitative ablation results for the $4096 \times 4096$ resolution are presented in Fig. 14, Fig. 15, Fig. 16, Fig. 17, and Fig. 18. Based on these results, it can be concluded that the basic numerical settings used in this experiment represent the optimal solution.

In Eq.7 and Eq.9 of the main text, $\omega_{\min}$ and $T_{\max}$ are fixed and thus excluded from ablation. The value of $N$ is kept consistent across both equations. For the $2048 \times 2048$ resolution setting, with $N$ set to 2, variations in $M_T$ and $M_\omega$ have minimal impact on performance. Therefore, only $N$, $\omega_{\max}$, and $T_{\min}$ are subject to ablation. The corresponding quantitative ablation results are shown in Fig. 11, Fig. 12, and Fig. 13. For the $4096 \times 4096$ resolution setting, we ablate $N$, $\omega_{\max}$, $T_{\min}$, $M_T$, and $M_\omega$. The corresponding results are presented in Fig. 14, Fig. 15, Fig.16, Fig.17, and Fig. 18. Based on these findings, we conclude that the default numerical settings used in our experiments yield the optimal performance.

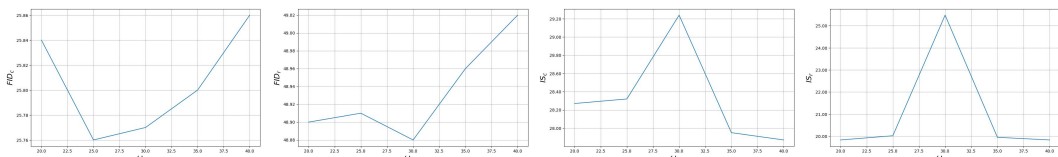

Figure 11: The image illustrates the ablation study of $\omega_{\max}$ in Eq.9 of the main text for the $2048 \times 2048$ resolution setting. The values of $\omega_{\max}$ range over $20, 25, 30, 35, 40$.

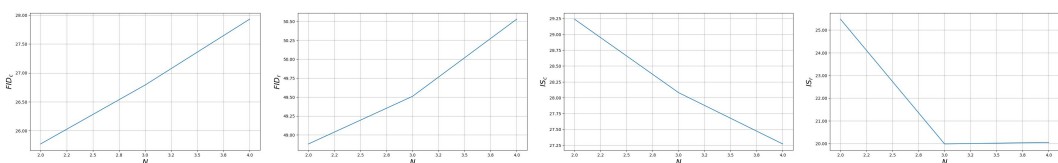

Figure 12: The image illustrates the ablation study of $N$ in Eq.7 and Eq.9 of the main text for the $2048 \times 2048$ resolution setting. The values of $N$ range over $2, 3, 4$.

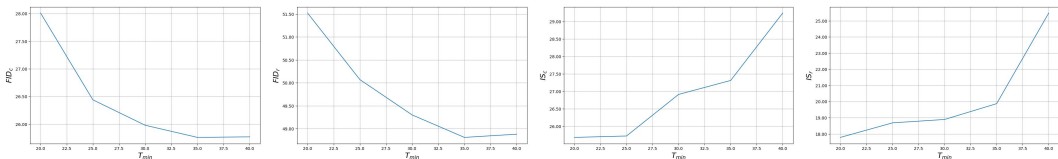

Figure 13: The image illustrates the ablation study of $T_{\min}$ in Eq.7 of the main text for the $2048 \times 2048$ resolution setting. The values of $T_{\min}$ range over $20, 25, 30, 35, 40$.

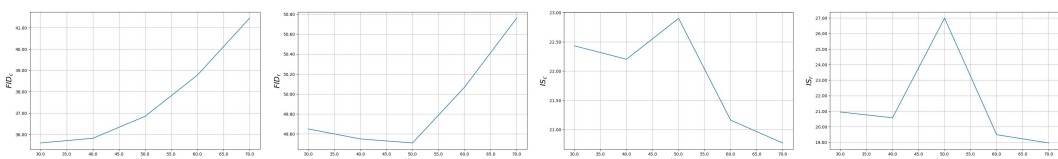

Figure 14: The image illustrates the ablation study of $\omega_{\max}$ in Eq.9 of the main text for the $4096 \times 4096$ resolution setting. The values of $\omega_{\max}$ range over $30, 40, 50, 60, 70$.

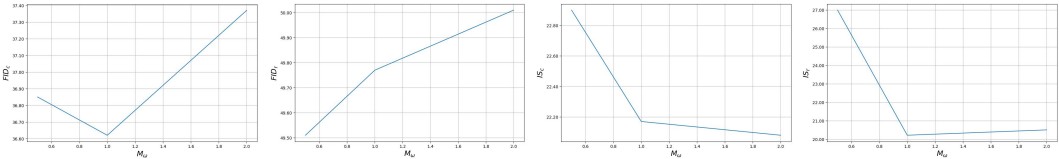

Figure 15: The image illustrates the ablation study of $M_\omega$ in Eq.9 of the main text for the $4096 \times 4096$ resolution setting. The values of $M_\omega$ range over $0.5, 1, 2$.

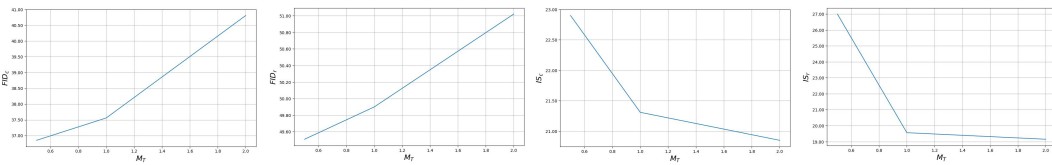

Figure 16: The image illustrates the ablation study of $M_T$ in Eq.7 of the main text for the $4096 \times 4096$ resolution setting. The values of $M_T$ range over $0.5, 1, 2$.

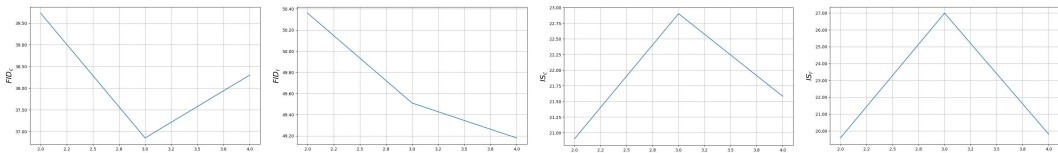

Figure 17: The image illustrates the ablation study of $N$ in Eq.7 and Eq.9 of the main text for the $4096 \times 4096$ resolution setting. The values of $N$ range over $2, 3, 4$.

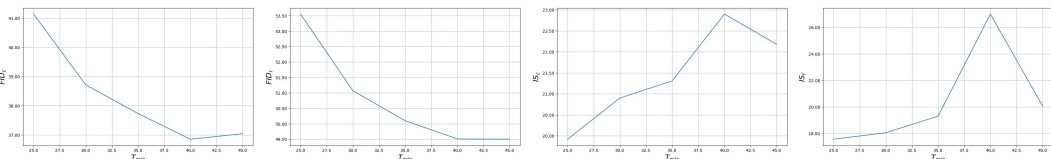

Figure 18: The image illustrates the ablation study of $T_{\min}$ in Eq.7 of the main text for the $4096 \times 4096$ resolution setting. The values of $T_{\min}$ range over $25, 30, 35, 40, 45$.

### A.9 HYPERPARAMETER DETAILS AND QUANTITATIVE RESULTS FOR APPLYING *RectifiedHR* TO APPLICATIONS

**The combination of *RectifiedHR* and WAN.** *RectifiedHR* can be directly applied to video diffusion models such as WAN (Wang et al., 2025). The officially supported maximum resolution for WAN 1.3B is $480 \times 832$ over 81 frames. Our goal is to generate videos at $960 \times 1664$ resolution using WAN 1.3B on an NVIDIA A800 GPU. The direct inference baseline refers to generating a $960 \times 1664$ resolution video directly using WAN 1.3B. In contrast, *WAN+RectifiedHR* refers to using *RectifiedHR* to generate the same-resolution video. The selected hyperparameters in Eq.7 and Eq.9 of the main text are: $N = 2$, $\omega_{\max} = 10$, $\omega_{\min} = 5$, $T_{\min} = 30$, $T_{\max} = 50$, $M_T = 1$, and $M_\omega = 1$. Our quantitative experimental details follow (Chen et al., 2023a) on 40 videos.

**The combination of *RectifiedHR* and OIR.** *RectifiedHR* can also be applied to image editing tasks. We employ SDXL as the base model and randomly select several high-resolution images from the OIR-Bench (Yang et al., 2023) dataset for qualitative comparison. Specifically, we compare two approaches: (1) direct single-object editing using OIR (Yang et al., 2023), and (2) OIR combined with *RectifiedHR*. While the OIR baseline directly edits high-resolution images, the combined method first downsamples the input to $1024 \times 1024$, performs editing via the OIR pipeline, and then applies *RectifiedHR* during the denoising phase to restore fine-grained image details. For the $2048 \times 2048$ resolution setting, the hyperparameters in Eq.7 and Eq.9 of the main text are: $N = 2$, $\omega_{\max} = 30$, $\omega_{\min} = 5$, $T_{\min} = 40$, $T_{\max} = 50$, $M_T = 1$, and $M_\omega = 1$. For the $3072 \times 3072$ resolution setting, the hyperparameters are: $N = 3$, $\omega_{\max} = 40$, $\omega_{\min} = 5$, $T_{\min} = 40$, $T_{\max} = 50$, $M_T = 1$, and $M_\omega = 1$.

**The combination of *RectifiedHR* and DreamBooth.** *RectifiedHR* can be directly adapted to various customization methods, where it is seamlessly integrated into DreamBooth without modifying any of the training logic of DreamBooth (Ruiz et al., 2023a). The base model for the experiment is SD1.4, which supports a native resolution of $512 \times 512$ and a target resolution of $1536 \times 1536$. The hyperparameters selected in Eq.7 and Eq.9 of the main text are as follows: $N$ is 3, $\omega_{\max}$ is 30, $\omega_{\min}$ is 5, $T_{\min}$ is 40, $T_{\max}$ is 50, $M_T$ is 1, and $M_\omega$ is 1. Furthermore, as demonstrated in Tab. 5, we conduct a quantitative comparison between the *RectifiedHR* and direct inference, using the DreamBooth dataset for testing. The test metrics and process were fully aligned with the methodology in (Ruiz et al., 2023a). It can be observed that *RectifiedHR* outperforms direct inference in terms of quantitative metrics for high-resolution customization generation.

*RectifiedHR* can be directly adapted to various customization methods and is seamlessly integrated into DreamBooth (Ruiz et al., 2023a) without modifying any part of its training logic. The base model used in this experiment is SD1.4, which natively supports a resolution of $512 \times 512$, with the target resolution set to $1536 \times 1536$. The selected hyperparameters in Eq.7 and Eq.9 of the main text are as follows: $N = 3$, $\omega_{\max} = 30$, $\omega_{\min} = 5$, $T_{\min} = 40$, $T_{\max} = 50$, $M_T = 1$, and $M_\omega = 1$. Furthermore, as shown in Tab.5, we conduct a quantitative comparison between

*RectifiedHR* and direct inference using the DreamBooth dataset for evaluation. The test metrics and protocol are fully aligned with the methodology described in (Ruiz et al., 2023a). The results demonstrate that *RectifiedHR* outperforms direct inference in terms of quantitative metrics for high-resolution customization generation.

| Direct Inference | DINO ↑ | CLIP-I ↑ | CLIP-T ↑ |
|---|---|---|---|
| DreamBooth + RectifiedHR | **0.625** | **0.761** | **0.249** |
| DreamBooth | 0.400 | 0.673 | 0.220 |

Table 5: Quantitative comparison results between *RectifiedHR* and direct inference after Dream-Booth training. The evaluation is conducted on a scene with a resolution of $1536 \times 1536$.

**The combination of *RectifiedHR* and ControlNet.** Our method can be seamlessly integrated with ControlNet (Zhang et al., 2023a) to operate directly during the inference stage, enabling image generation conditioned on various control signals while simultaneously enhancing its ability to produce high-resolution outputs. The base model used is SDXL. The selected hyperparameters in Eq.7 and Eq.9 of the main text are: $N = 3$, $\omega_{\max} = 40$, $\omega_{\min} = 5$, $T_{\min} = 40$, $T_{\max} = 50$, $M_T = 1$, and $M_\omega = 1$.

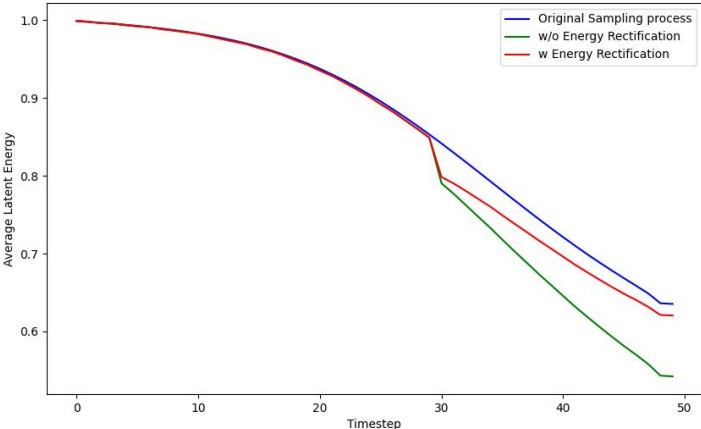

Figure 19: Visualization of the average latent energy curve following energy rectification.

## A.10 VISUALIZATION OF THE ENERGY RECTIFICATION CURVE

To better visualize the average latent energy during the energy rectification process, we plot the corrected energy curves. We randomly select 100 prompts from LAION-5B for the experiments. As shown in Fig. 19, the blue line represents the energy curve at a resolution of $1024 \times 1024$. For the $2048 \times 2048$ resolution setting, we use the following hyperparameters: $T_{\min} = 30$, $T_{\max} = 50$, $N = 2$, $\omega_{\min} = 5$, $\omega_{\max} = 30$, $M_T = 1$, and $M_\omega = 1$. The red line corresponds to our method with energy rectification for generating $2048 \times 2048$ resolution images, while the green line shows the result of our method without the energy rectification module. It can be observed that energy rectification effectively compensates for energy decay.

## A.11 IMPLEMENTATION DETAILS

Although a limited number of samples may lead to lower values for metrics such as FID (Heusel et al., 2017), we follow prior protocols and randomly select 1,000 prompts from LAION-5B (Schuhmann et al., 2022) for text-to-image generation. Evaluations are conducted using 50 inference steps, empty negative prompts, and fixed random seeds.

We employ four widely used quantitative metrics: Fréchet Inception Distance (FID) (Heusel et al., 2017), Kernel Inception Distance (KID) (Bińkowski et al., 2018), Inception Score (IS) (Salimans et al., 2016), and CLIP Score (Radford et al., 2021). FID and KID are computed using `pytorch-fid`, while CLIP Score and IS are computed using `torchmetrics`. The subscript $r$ refers to resizing high-resolution images to $299 \times 299$ before evaluation, whereas the subscript $c$ indicates that 10 patches of size $1024 \times 1024$ are randomly cropped from each generated high-resolution image and then resized to $299 \times 299$ for evaluation. Specifically, $\text{FID}_r$, $\text{KID}_r$, and $\text{IS}_r$ require resizing images to $299 \times 299$. However, such an evaluation is not ideal for high-resolution image generation. Following prior works (Du et al., 2024; Lin et al., 2025), we randomly crop 10 patches of size $1024 \times 1024$ from each generated high-resolution image to compute $\text{FID}_s$, $\text{KID}_c$, and $\text{IS}_c$.

## A.12 MORE VIDEO RESULTS

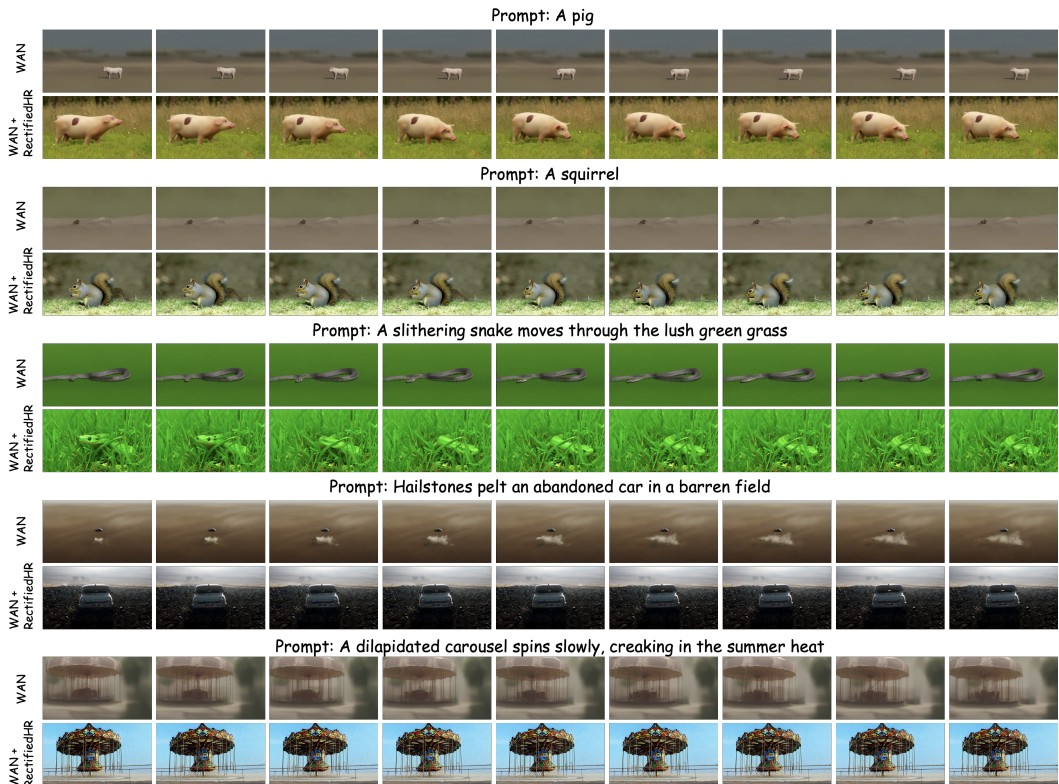

Figure 20: More video results

## A.13 MORE IMAGE RESULTS

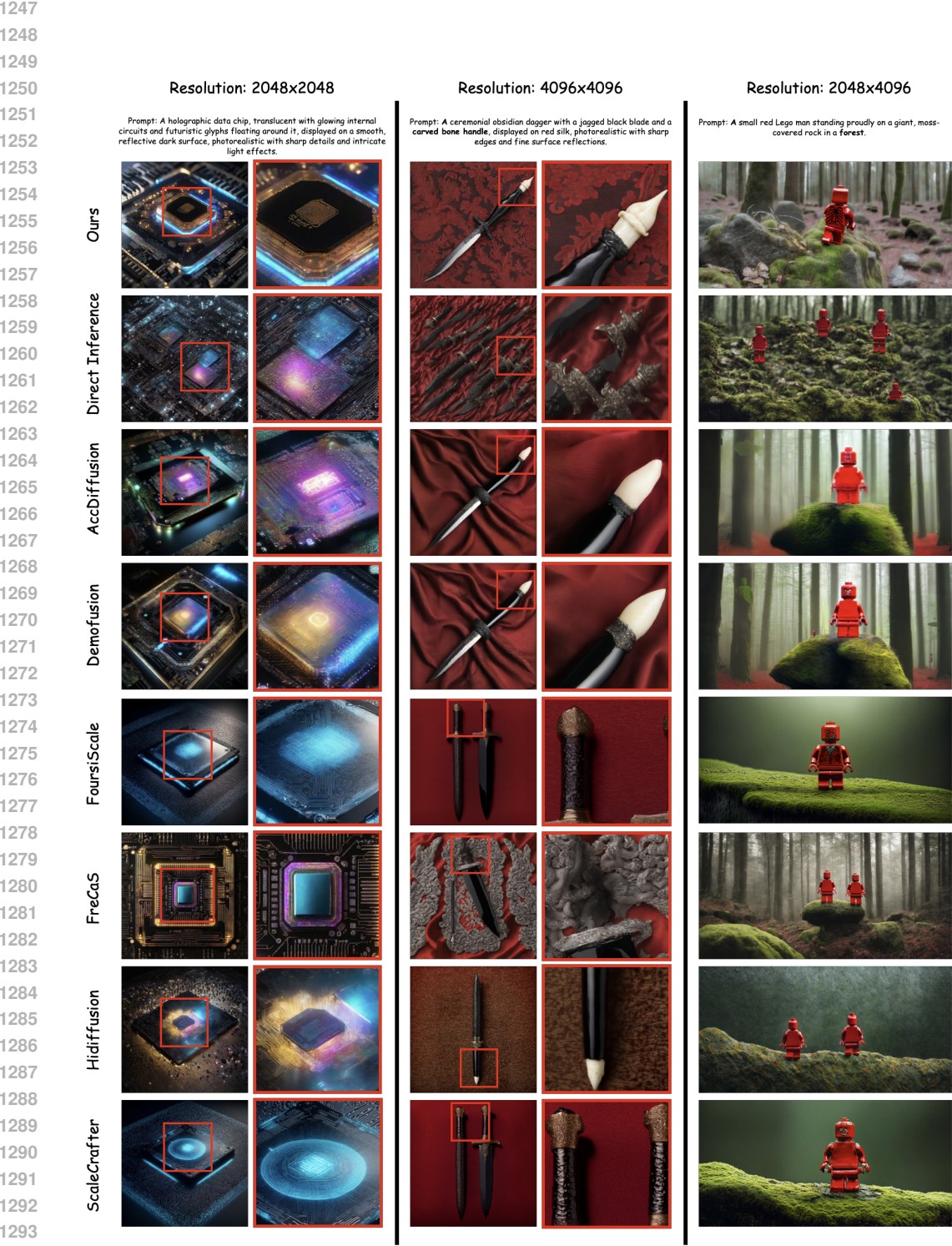

Figure 21: More image results

## A.14    USER STUDY DETAILS

*1. **Note**: This questionnaire may take about 10 minutes.

We generated six high–resolution images (2048 × 2048) using different AI techniques, all based on the same prompt: *"An opulent **crystal chandelier** with hundreds of sparkling glass prisms reflecting warm candlelight, suspended in a grand ballroom with ornate golden ceiling details, photorealistic with intricate refractions and glowing highlights."*

Please choose the image that you think is the **best overall**. When making your choice, you can consider the following aspects:
**Prompt adherence**: How well the image matches the description in the prompt
**Detail and clarity**: The richness of textures and whether the image quality is clear.
**Consistency**: Whether shapes, perspective, and lighting are coherent
**Overall aesthetics**: Color harmony, composition, and visual appeal

For each image, we have also highlighted some detail regions for your reference. However, please remember **not to focus solely on** the highlighted areas.

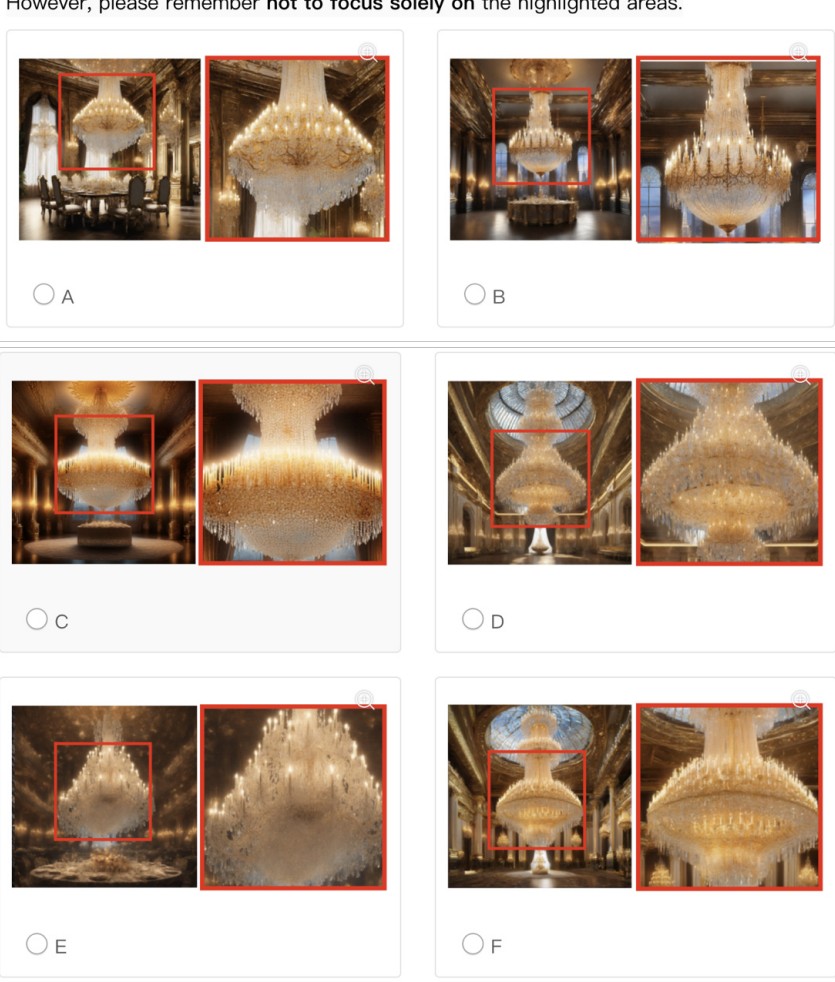

Figure 22: The interface of one question in the user study

We conducted a user study to further demonstrate the effectiveness of our method. We selected 15 images in total, evenly distributed across three resolutions: $2048 \times 2048$, $4096 \times 4096$, and $2048 \times 4096$ (five images per resolution). 30 participants were involved in the study, where they

were asked to evaluate the images provided and identify the best. The questionnaire is designed on the `https://www.wjx.cn/` platform. The interface of the questionnaire is shown in Fig. 22.

The baselines in this study are consistent with those in Sec. A.2, except for direct inference and DemoFusion. Direct inference was excluded because most of its generated images exhibited severe global distortions. The outputs of AccDiffusion and DemoFusion are highly similar under a fixed random seed. As (Lin et al., 2025) has quantitatively demonstrated the superiority of AccDiffusion, we retained AccDiffusion solely for conciseness in this study.

Fig. 23 shows the results of the user study. Our method (RectifiedHR) received 32.2% of the total votes, significantly exceeding the other competing methods. The second most selected method, FreCaS, accounted for only 16.2%, which is approximately half of RectifiedHR's proportion. The remaining methods, including AccDiffusion (13.8%), ScaleCrafter (13.6%), HiDiffusion (12.7%), and FouriScale (11.5%), received relatively lower proportions of the total votes. These results demonstrate that more users are inclined to identify RectifiedHR as the best compared to existing approaches, validating the effectiveness of our method in subjective evaluation.

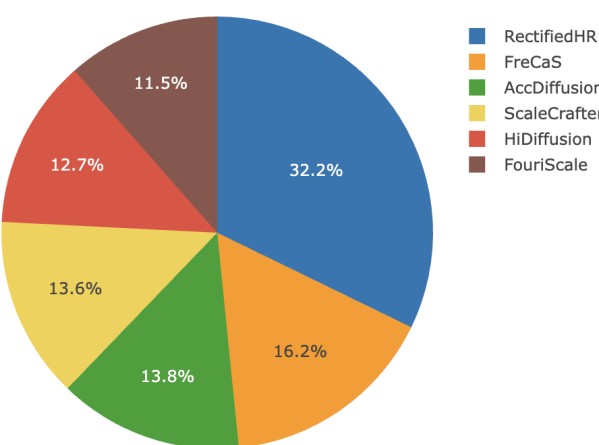

Figure 23: The results of the user study

