# OpenReview forum: "Efficient Training-Free High-Resolution Synthesis with Energy Rectification in Diffusion Models"
_ICLR.cc/2026/Conference — ICLR 2026 Conference Withdrawn Submission_

### Official Review · Reviewer_5YHq · 2025-10-29

**Soundness:** 3
**Presentation:** 4
**Contribution:** 2
**Rating:** 4
**Confidence:** 5

**Summary:**

This paper presents a train-free method for high-resolution image generation. It defines a "latent energy" metric based on variance and observes that this energy decreases over the denoising process. The authors also note that classifier-free guidance (CFG) influences the latent energy. Subsequently, they propose a sampling strategy that involves upsampling the predicted x₀, resizing it, and projecting it back into the latent space, followed by noise addition. Theoretical equations are provided to establish a connection between energy rectification and signal-to-noise ratio (SNR).

**Strengths:**

+ The discovery of denoising-process-related statistics, such as latent energy, is intriguing.
+ The proposed noise refresh strategy helps reduce blurriness.
+ The method achieves superior performance compared to other competing approaches.

**Weaknesses:**

- Latent energy is merely a statistical measure associated with blur, but it fails to capture issues like texture repetition (e.g., in astronaut clothing, water, and fog in Figure 1). Variance is not an ideal metric for evaluating high-resolution generation quality, as the overall structure largely depends on pixel-space upsampling.
- The noise refresh strategy resembles SDEdit, a commonly used technique for enhancing image clarity (e.g., in I2VGen), and thus lacks novelty.
- The overall visual results are unsatisfactory, exhibiting problems such as texture repetition and blurriness. Such a train-free approach may not address the core challenges of high-resolution generation.
- The assumption in Equation (10) may be invalid, as ε(·) is a function of xₜ.

**Questions:**

Please refer to the strengths and weaknesses

---

### Official Review · Reviewer_MXQU · 2025-10-30

**Soundness:** 2
**Presentation:** 3
**Contribution:** 1
**Rating:** 2
**Confidence:** 4

**Summary:**

This paper presents RectifiedHR, a training-free approach for high-resolution image synthesis in diffusion models. The authors claim that their method solves the issue of image blurriness during high-resolution generation by introducing a noise refresh strategy and a "energy rectification" mechanism. The paper further proposes tuning the classifier-free guidance (CFG) hyperparameter at each step to compensate for energy decay. Extensive experiments are conducted, comparing RectifiedHR to existing methods and demonstrating its speed and image quality improvements.

**Strengths:**

1. The paper is generally well-written, and the methodology is presented clearly. The authors successfully introduce new concepts like energy rectification and noise refresh, which are relatively easy to understand within the context of diffusion models.
2. The proposed method is relatively efficient compared to existing approaches.
3. The experimental results demonstrate that RectifiedHR outperforms several state-of-the-art (SOTA) methods in terms of speed and image quality. The method also adapts well to various applications, such as image editing, customized generation, and video synthesis.

**Weaknesses:**

1. The core contribution centers on “energy rectification,” posited as essential for reducing image blurriness at high resolutions. However, the link between energy levels and image quality is not convincingly justified. The concept of “energy decay” as the cause of blurriness lacks a solid theoretical explanation, and the argument for why this decay specifically degrades image fidelity remains unsubstantiated.
2. The noise refresh technique itself is not conceptually novel and resembles existing resolution-refreshing or re-noising operations seen in prior training-free diffusion approaches, for example in [4].
3. While the paper discusses the relationship between energy decay and the loss of high-frequency components, it does not clearly establish why or how this decay directly harms high-frequency details. This ambiguity weakens the claimed novelty of the energy rectification module.
4. The adaptive tuning of the CFG parameter across timesteps is already a well-established practice in diffusion research. Prior studies—such as Kynkäänniemi et al. [1], Malarz (2024) [2], and Castillo [3]—have explored step-wise and adaptive CFG tuning extensively. Yet, the paper provides limited discussion or comparison with these works.
5. The effectiveness of RectifiedHR appears to be highly sensitive to hyperparameter choices, particularly the timing and magnitude of CFG adjustments, which raises concerns about generalizability.
6. Although energy rectification is proposed as a novel mechanism to counter energy decay, the paper does not adequately differentiate it from related ideas such as SNR correction or existing CFG optimization techniques. A deeper comparative analysis would clarify whether the observed improvements arise from genuinely new principles or simply from parameter tuning.

[1] Tuomas Kynkäänniemi, et al. Applying Guidance in a Limited Interval Improves Sample and Distribution Quality in Diffusion Models. NeurIPS 2024.

[2] Dawid Malarz, et al. Classifier-free Guidance with Adaptive Scaling.

[3] Angela Castillo, et al. Adaptive guidance: Training-free acceleration of conditional diffusion models. AAAI 2025.

[4] Wongi Jeong, et al. Upsample What Matters: Region-Adaptive Latent Sampling for Accelerated Diffusion Transformers

**Questions:**

See above.

---

### Official Review · Reviewer_UnSk · 2025-10-30

**Soundness:** 3
**Presentation:** 2
**Contribution:** 3
**Rating:** 6
**Confidence:** 4

**Summary:**

This paper, proposes RectifiedHR, a training-free high-resolution image generation approach aimed at efficiency. The work makes two key contributions: a noise refresh strategy and a tuning strategy for the classifier-free guidance hyperparameter, which is shown to improve generation performance. The approach is tested with different models and tasks. Promising results are shown on the selected datasets.

**Strengths:**

- Paper is well-structured, with easy to follow sections and sufficient ablation studies for the main components
- Good results, particularly when accounting for the speed gains
- The two proposed components, noise refresh and energy rectification, are interesting
- Good insights

**Weaknesses:**

- Missing comparison with some sota methods (e.g: FAM Diffusion: Frequency and Attention Modulation for High-Resolution
Image Generation with Stable Diffusion, Yang et el, CVPR 2025)
- A typical failure case for zero-shot high resolution image generation method is that they are prune to various artifacts, especially as the resolution grows (repetitive patterns, misplaced textures etc). An evaluation for such failure modes will be a good addition as most of the metrics will not capture this. In addition to some quantitative evaluation, some qualitative examples perhaps showing a few random prompts and their output at different resolutions would be helpful.
- The accuracy gains compared with DiffuseHigh are marginal

**Questions:**

- What is the dataset used to report the main results from the paper? Is the result consistent across different datasets?
- I notice in the abstract that there is a 8K image, but I couldn't see a quantitative evaluation for this. Are there any numerical results?
- How does the performance of the proposed approach and the other sota methods varies when changing the number of denoising steps from 50? How does this vary with increased resolution?

---

### Official Review · Reviewer_bSGM · 2025-10-31

**Soundness:** 3
**Presentation:** 3
**Contribution:** 2
**Rating:** 6
**Confidence:** 4

**Summary:**

The authors propose a training-free method to enable video generators for high resolution generation. The method, called RectifiedHR, adopts a noise refresh strategy to achieve it. The authors observe the energy decay phenomenon in high-resolution synthesis. To resolve it, RectifiedHR incorporates the average latent energy analysis and a CFG hyperparameter tuning strategy. Experiments show RectifiedHR can achieve promising high-resolution video generation without training.

**Strengths:**

1. The method does not require additional training. And achieves the best result compared to the baselines.
2. Extensive ablation experiments show the effectiveness of the technical components.
3. This method is succesfully appied to a variety of base models and visual generation tasks.
4. Compared to other training-free high resolution generation methods. RectifiedHR is very fast and computation friendly.

**Weaknesses:**

1. A quantitative comparison in video generation is lacking. Since DemoFusion is also a training-free method. The authors can try to compare to it on a video generation benchmark.
2. The noise refresh strategy is a commonly used method. The authors should emphasis the difference of it in RectifiedHR compared to PyramidDiffusion and RelayDiffusion[2].
3. Have you tried other CFG weight scaling strategies? Is Eqn.9 supported by any theoretical analysis? If not, can you compare it with some possible choices? (e.g., linear interpolation, softmax function.)

[1] https://arxiv.org/abs/2208.01864
[2] https://arxiv.org/abs/2309.03350

**Questions:**

Please see weaknesses.

---

### Note · Authors · 2025-11-12

I have read and agree with the venue's withdrawal policy on behalf of myself and my co-authors.